# Effect of Ho Addition on the Glass-Forming Ability and Crystallization Behaviors of Zr_54_Cu_29_Al_10_Ni_7_ Bulk Metallic Glass

**DOI:** 10.3390/ma15072516

**Published:** 2022-03-29

**Authors:** Shuaidan Lu, Xiaoxiao Li, Xiaoyu Liang, Jiahua He, Wenting Shao, Kuanhe Li, Jian Chen

**Affiliations:** 1School of Materials Science and Chemical Engineering, Xi’an Technological University, Xi’an 710021, China; lushuaidan@hotmail.com (S.L.); lixiaoxiao_stu@163.com (X.L.); liangxiaoyu@st.xatu.edu.cn (X.L.); hejiahua@xatu.edu.cn (J.H.); shaowenting@xatu.edu.cn (W.S.); 2School of Metallurgy, Northeastern University, Shenyang 110819, China; kevinlee579@outlook.com

**Keywords:** Zr-based metallic glasses, crystallization kinetic, holmium, activation energy

## Abstract

The effect of holmium (Ho) addition on the glass-forming ability (GFA) and crystallization behaviors of Zr_54_Cu_29_Al_10_Ni_7_ bulk metallic glass (BMGs) were studied by employing differential scanning calorimetry (DSC), X-ray diffraction (XRD), and transmission electron microscopy (TEM). The characteristic temperatures and activation energies of crystallization were obtained from DSC data. Classical kinetic modes were used to evaluate the crystallization processes of Zr_54_Cu_29_Al_10_Ni_7_ and Zr_48_Cu_29_Ni_7_Al_10_Ho_6_ BMGs. The results showed that Ho addition reduces the activation energy in the original crystallization period of Zr-based BMG and improves the nucleation, which is due to the formation of simpler compounds, such as CuZr_2_, Cu_2_Ho, and Al_3_Zr_5_.

## 1. Introduction

Zr-based bulk metallic glasses (BMGs) are considered to possess a widely utilized future as excellent structural materials in consumer electronics for their extraordinary fracture strengths, corrosion, and wear resistance [1,2,3,4]. In addition, the good glass-forming ability (GFA) of Zr-based BMGs allows thermoforming production above their glass transition temperature (*T_g_*) [5,6,7,8,9,10]. However, the plasticity of Zr-based BMGs needs some improvement for commercial applications [11,12]. Recent investigations have reported that the plasticity of BMGs is enhanced in an amorphous composite by separating out micro-nano sized grains, which blocks the development of initial shear bands and forms abundant small-scale shear bands [13,14,15]. It is well known that crystallization of glasses is a very effective way of getting bulk nanocrystalline materials and crystal/glass composite [16,17,18]. Therefore, it is necessary to study the crystallization behavior of BMGs for the controllable preparation of crystal/glass composite.

Microalloying is widely used to improve the properties of alloys, especially those of metallic glasses [19,20,21,22,23,24,25]. Due to special chemical properties, rare earth elements are usually used as beneficial alloys additives to obtain better performance and are called “industrial monosodium glutamate” [26,27,28,29]. In our previous work, 1 at.% addition of rare earth Y was found to improve the GFA of Zr-based BMGs effectively and change the first crystallization phase of Zr_53.8_Cu_29.1_Ni_7.3_Al_9.8_ BMG from Cu_10_Zr_7_ to CuZr_2_, which is beneficial to controlling the crystallization process and preparing bulk nanocrystalline alloys [30]. Microalloying of rare earth holmium (Ho) element was investigated to show beneficial effects on the GFA of Fe-based and Zr-based metallic glasses [26,27,28,29,31,32]. However, the addition of Ho in the literature is minimal and displayed limited understanding of microalloying effects of Ho [26,27,28,29,32]. In addition, the influence of Ho addition on the crystallization of metallic glasses has not been studied yet.

In this work, metallic glasses with composition of Zr_54−*x*_Cu_29_Ni_7_Al_10_Ho*_x_* (*x* = 0, 1, 2, 4, 6) were prepared and the influence of Ho addition on the GFA and crystallization behavior was systematically investigated. The characteristic temperatures and corresponding activation energies were calculated. Crystallization kinetics equations were fitted to evaluate the crystallization processes, and the phase changes were researched by employing X-ray diffraction (XRD) and transmission electron microscopy (TEM).

## 2. Materials and Methods

The Zr_54−*x*_Cu_29_Ni_7_Al_10_Ho*_x_* (*x* = 0, 1, 2, 4, 6) master alloy ingots were prepared with purity metals (99.99%), and designated as Zr54, Zr53Ho1, Zr52Ho2, Zr50Ho4, and Zr48Ho6. Each prepared ingot was remelted five times by electric arc melting. The Ti ingot was melted to ensure an oxygen-free atmosphere. Later, as-cast rods (diameter of 3 mm and length of 80 mm) were obtained by suction casting into a water-cooled copper mold following induction smelting. The amorphous structure of these alloys was identified by X-ray diffraction (XRD, Bruker D8, Bruker AXS, Karlsruhe, Germany, Cu-Kα radiation). The crystallization peaks were measured by differential scanning calorimetry (DSC, STA449F3, Netzsch, Selb, Gemany). The DSC tests of BMGs with different Ho additions were carried at a fixed heating rate of 10 K/min and the non-isothermal tests were carried at various heating rates of 5, 10, 20, 30, 40, and 50 K/min. The crystallization experiments in different duration were carried at 500 °C, and the corresponding phase changes were investigated by XRD and TEM (G20, FEI, Hillsboro, OR, USA). TEM samples were prepared by ion milling with the ion-beam voltage of 3.5 keV under the temperature of −40 °C to avoid structural change.

## 3. Results and Discussion

### 3.1. Structure and GFA of Metallic Glasses

Figure 1 is the XRD patterns of as cast Zr_54−*x*_Cu_29_Ni_7_Al_10_Ho*_x_* (*x* = 0, 1, 2, 4, 6) BMGs. All the patterns display the broad diffraction peaks in the range of 2θ = 30–50°, and there is no sharp peak observed in the patterns, implying the amorphous structure. To study the influence of Ho alloying on the GFA of Zr_54_Cu_29_Ni_7_Al_10_ BMG, DSC tests were conducted and the resulting curves are displayed in Figure 2. The glass transition temperature (*T_g_*) and initial crystallization temperature (*T_x_*) can be obtained from the DSC curves and summarized in Table 1. The super-cooled liquid region Δ*T_x_* (=*T_x_* − *T_g_*) can also be calculated and listed in Table 1. It can be found that *T_g_* and *T_x_* both decrease with the increasing Ho addition. However, the drop of *T_g_* is larger than that of *T_x_*, leading to an increase of Δ*T_x_*. The largest Δ*T_x_* value of Zr48Ho6 BMG implies the greatest GFA. Based on the results of GFA, Zr_48_Ho_6_ BMG was selected to carry the further studies about crystallization kinetic of Zr_54_Cu_29_Ni_7_Al_10_ BMG.

### 3.2. Non-Isothermal Crystallization Kinetics

Figure 3 shows the DSC traces of Zr54 and Zr48Ho6 BMGs achieved at various heating rates. All the DSC traces show the glass transition characters and exothermic peaks of crystallization. The characteristic temperatures are pointed in Figure 3 and listed in Table 2. As seen in Figure 3, the exothermic peaks move to higher temperatures with the raising heating rate. The characteristic temperatures also become higher at the faster heating rate, indicating an obvious correlation between crystallization and the heating rate, because the crystallization consists of the nucleation and the growth of grains, which are the thermally activated process. In addition, it can also be found in Table 2 that *T_g_* increases more slowly than *T_x_*, leading to the improved Δ*T_x_*.

The volume fraction of crystallization, *x*, could be calculated using DSC results by taking the line integral of the crystallization peak. When temperature *T* is fixed, the corresponding *x* can be obtained by *x* = *S_T_*/*S_p_*, where *S_p_* represents the total area of exothermic peak, and *S_T_* represents the area below the peak curve between the initial temperature and the fixed temperature *T*. The obtained crystallized fraction *x* is plotted in Figure 4 and displays the classic S-shape, which illustrates the process of non-isothermal crystallization [33,34,35].

### 3.3. Effective Activation Energy

The effective activation energy, *E*, can be calculated from DSC results by using a Kissinger Equation (1) [36] and Ozawa Equation (2) [37]:(1)$$\ln\left(\beta /T^{2}\right)=-E/RT+C$$
(2)lnβ=−E/RT+C

In the equations, *β* is the rate of raising temperature, *T* is characteristic temperature, and R is gas constant. The Kissinger results ln(*β*/*T*^2^) and Ozawa results ln(*β*) against 1000/R*T* are shown as scattered points in Figure 5. Each set of data is linearly fitted, and the corresponding results are also displayed in Figure 5. According to the Kissinger and Ozawa equations above, the slope obtained from linear fitting is *E*/R, and the corresponding activation energies for Zr_54_ and Zr_48_Ho_6_ BMGs are calculated from the slopes and summarized in Table 3. As seen in Table 3, the values obtained by Ozawa equation are in good agreement with those obtained by a Kissinger equation, and are merely a little bit higher.

It is known that *E_g_*, *E_x_*, and *E_p_* represent the energy barriers to overcome for grass transition, nucleation, and growth of crystal. It can be found in Table 3 that *E_g_* is much greater than others, implying the greater energy barrier to overcome for atomic rearrangement. Comparing with *E_g_* of Zr48Ho6 (488.87 kJ/mol and 500.04 kJ/mol), *E_g_* of Zr54 obtained by Kissinger and Ozawa methods are 1525.07 kJ/mol and 1537.03 kJ/mol, and much larger, implying a more difficult atomic rearrangement. It can also be found that *E_x_* is larger than *E_p_* for Zr54 BMG, indicating the hard nucleation and easy grain growth. However, *E_x_* is smaller than *E_p_* for Zr48Ho6 BMG, implying the easy nucleation and hard growth. The transformation should be attributed to the changes of crystallization products caused by Ho alloying.

### 3.4. Effective Activation Energy

The activation energy obtained from *T_p_* just roughly evaluates the crystallization process. However, as the nucleation and growth of metallic glass are complicated, the activation energy over the total crystallization process is not fixed. The local activation energy against crystallized fraction *x*, *E_c_*(*x*), was proposed to evaluate the whole crystallization process. It can be calculated by the Doyle–Ozawa isoconversional equation (seen in Equation (3)) [38,39]. The results are displayed in Figure 6 and show two quite different tendencies of *E*_c_(*x*) for two BMGs. *E*_c_(*x*) for Zr54 reaches a peak point of 660.40 kJ/mol in the original period and next declines immediately to 600.84 kJ/mol at *x* = 1%; thereafter, it unceasingly declines with the increasing *x* and lastly finishes at 174.06 kJ/mol (*x* = 99.5%). Meanwhile, *E*_c_(*x*) for Zr48Ho6 drops from the peak point of 412.17 kJ/mol at the original point to 385.50 kJ/mol at *x* reaches 1%; next, *E*_c_(*x*) declines continuously in the following process and ends up with 222.92 kJ/mol at *x* = 99.4%. The mean activation energy for Zr54 and Zr48Ho6 BMGs during the whole crystallization process are 366.43 kJ/mol and 332.33 kJ/mol, respectively, less than the values calculated by the Kissinger and Ozawa equations. Similar results are reported in the crystallization behaviors of some other metallic glasses [38,39,40]. Comparing with Zr54 BMG, *E*_c_(*x*) for Zr48Ho6 is smaller in the original period and larger in the second half process, implying an easier nucleation and more difficult development. These are in good agreement with the results obtained from Kissinger and Ozawa methods:(3)$$\ln\beta =-1.0516E_{c}\left(x\right)/RT+C$$

### 3.5. Crystallization Mechanism

The solid-state reaction can be described by the following equation:(4)dx/dt=k(T)f(x)
where *x* is the crystallized fraction, *t* is the duration, *T* is the reaction temperature, *f*(*x*) is set as an expression describing the crystallization process, and *k*(*T*) is a constant dependent on temperature and can be obtained by the Arrhenius model [41]:(5)$$k\left(T\right)=A\;\exp\left(-E_{c}/RT\right),$$
where *A* is frequency factor, and *E*_c_ is crystallization activation energy. Combining Equation (4) with Equation (5), the following equation can be obtained:(6)$$\ln\left(dx/dt\right)+E_{c}/RT=\ln\left[Af\left(x\right)\right].$$

Based on Equation (6), the value of ln[*Af*(*x*)] can be obtained from crystallized fraction and activation energy of *T*_p_ (calculated by the Kissinger equation and seen in Table 3). Because the curves of crystallized fraction were varied at various heating rates, the mean values of ln[*Af*(*x*)] were taken to describe the crystallization and plotted as scatters (hollow circles) in Figure 7. The Johnson–Mehl–Avrami (JMA) mode and normal grain growth (NGG) mode in Table 4 were employed to evaluate the experimental data [42,43]. The fitted curves are in red and dotted lines, and the corresponding results are listed in Figure 7.

As seen in Figure 7a that the crystallization of Zr54 BMG firstly obeys the JMA-mode with *n* = 1.8075 for *x* < 0.16, then the crystallization transforms to obey two NGG-modes (*m* = 3.0311 and 2.4087). As seen in Figure 7b, the crystallization of Zr48Ho6 BMG starts with the JMA-mode with a greater *n* = 2.5154 for *x* < 0.21; later, the remaining crystallization follows two NGG straight lines with less *m* = 2.3204 and 1.9803, respectively.

As is known to all, the JMA mode describes the precipitation process from nucleation to growth, which commonly takes place in the original period of precipitation. The value of JMA exponent *n* for Zr54 BMG is <2.5, suggesting a growth in three dimensions and a lowering nucleation controlled by diffusion [43]. However, the JMA exponent value of Zr48Ho6 BMG is >2.5, implying a growth in three dimensions and rising nucleation controlled by diffusion. Comparing with Zr54 BMG, the greater JMA exponent of Zr48Ho6 BMG implies a faster nucleation rate. Moreover, the JMA-mode stage of Zr48Ho6 BMG keeps to *x* = 0.21 and is longer than that of Zr54 BMG (*x* = 0.16), suggesting the more nucleation in Zr48Ho6 BMG. This result is consistent with the less *E_x_* for Zr48Ho6 BMG achieved by the Kissinger method and the Doyle–Ozawa isoconversional method. The developed nucleation should be due to micro-alloying of rare earth, which raises the viscosity of super cooled liquid and lowers the driving force of nucleation [44].

In the follow-up period, the crystallization of Zr54 and Zr48Ho6 BMGs both follow the NGG-mode. Comparing with Zr54 BMG, the lesser NGG exponent of Zr48Ho6 BMG suggests decelerated growth of grain. It indicates that 6 at.% addition of Ho has the tardiness effect on the growth of grain and is beneficial to prepare bulk nanocrystalline alloys by controlling the crystallization process [45].

### 3.6. Structural Analysis of the Crystallized Alloys

To explain the different crystallization mechanisms of two BMGs, the structure of two alloys annealed at 500 °C was investigated by XRD, and the obtained spectra are analyzed in Figure 8. There is a large amount of CuZr_2_, Ni_10_Zr_7_, Cu_2_Ho, Al_3_Zr_5_, and Al_3_Zr were formed on Zr48Ho6 BMG annealed at 500 °C for 20 min, while Cu_10_Zr_7_ and few Al_2_Zr were detected on Zr54 BMG after 20 min annealing treatment. The results of 40 min annealing indicate that these primary grains grow up with the continuous annealing. It is obvious that the crystallization processes of two BMGs are essentially different. The formation of Cu_2_Ho on Zr48Ho6 BMG is attributed to the addition of Ho, which leads to the reduced Cu atoms combining with Zr atoms, and then causes the precipitation of CuZr_2_, Ni_10_Zr_7_, and Al_3_Zr_5_. The results indicate that Ho tends to combine with Cu rather than other elements. According to the mixing enthalpy between elements in Figure 9, the mixing enthalpies of Ni-Ho and Al-Ho are more negative than that of Cu-Ho, but less negative than those of Ni-Zr, Ni-Al, and Zr-Al. Ni will combine preferentially with Zr and Al, and Al will combine preferentially with Zr and Ni, rather than Ho. Therefore, Ho combined preferentially with Cu to form precipitates. Meanwhile, Zr-Cu, Zr-Ni, and Zr-Al also preferentially form the corresponding compounds, which is consistent with the XRD results.

Figure 10 exhibits the TEM photographs of Zr54 and Zr48Ho6 BMGs after 20 min of annealing at 500 °C, and the corresponding selected area electron diffraction (SAED) patterns are inset in the upper right corner. As shown in Figure 10a, there are scattered precipitates, roughly 3050 nm, that appear in Zr54 BMG. The SAED pattern shows many diffraction spots and indicates the formation of Cu_10_Zr_7_ and Al_2_Zr. The TEM image of Zr48Ho6 in Figure 10b also shows a well-ordered polycrystalline structure with immense amounts of precipitates. The SAED pattern indicates the existence of CuZr_2_, Ni_10_Zr_7_, and Cu_2_Ho phases. Compared with the XRD analysis of annealed alloys, it could be inferred that the crystallization mechanism of Zr54 BMG is changed by Ho addition. It has been found in the nucleation mechanism of Zr-Cu-Al BMGs, which could be divided into two categories: CuZr_2_-type and Cu_10_Zr_7_-type [46]. CuZr_2_-type represents the lower free energy barrier and easy nucleation, attributing to the simple space unit of CuZr_2_. A Cu_10_Zr_7_-type implies a more chemically and topologically disordered crystal structure, resulting in much harder nucleation. The similar result has also been found in our previous work about the influence of Y alloying on the crystallization behavior of Zr-based BMGs [30]. In this work, it is clear that Ho addition reduces the free energy barrier of nucleation and leads to the formation of simple phases, such as CuZr_2_ and Cu_2_Ho. This is in good agreement with the less activation energy *E_x_* and the improved nucleation in JMA mode obtained above. As the initial crystallized phase is mainly of Cu_10_Zr_7_, the crystallization of Zr54 BMG belongs to Cu_10_Zr_7_-type. Compared with the addition of Y, the alloying of Ho led to the different precipitates Cu_2_Ho, which resulted in the difference of crystallized products and the corresponding proportion [30]. Then, the nanocrystalline alloy with different structures and properties can be obtained.

## 4. Conclusions

The influences of Ho addition on the GFA and crystallization process of Zr_54_Cu_29_Al_10_Ni_7_ BMGs were studied. The results were concluded to be as follows:(1)The crystallization behavior of Zr54 BMG was changed by Ho addition. The non-isothermal DSC studies showed a good correlation between characteristic temperatures and heating rate.(2)The Kissinger, Ozawa, and Doyle–Ozawa isoconversional methods all provided less activation energy at the initial crystallization point of Zr48Ho6 BMG, indicating the much easier nucleation.(3)The crystallization of Zr54 and Zr48Ho6 BMG both firstly obeyed the JMA-mode and then changed to obey the NGG-mode. The larger JMA exponent of Zr48Ho6 BMG implies the improved nucleation. The smaller NGG exponents for the following crystallization of Zr48Ho6 BMG mean a slower growth rate of precipitates.(4)The XRD and TEM analyses indicated that the initial crystallized products of Zr48Ho6 BMG were simple compounds, such as CuZr_2_, Cu_2_Ho, and Al_3_Zr_5_, while the initial crystallization of Zr54 BMG was the formation of Cu_10_Zr_7_ with a complex structure.

## Figures and Tables

**Figure 1 materials-15-02516-f001:**
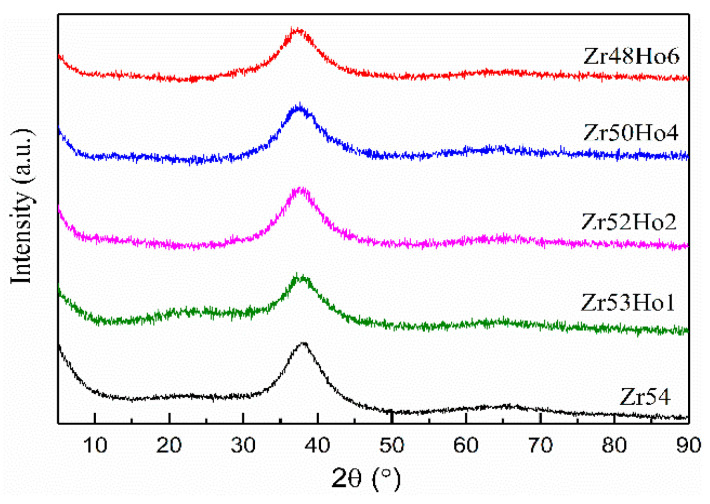
XRD patterns of Zr_54−*x*_Cu_29_Ni_7_Al_10_Ho*_x_* (*x* = 0, 1, 2, 4, 6) BMGs.

**Figure 2 materials-15-02516-f002:**
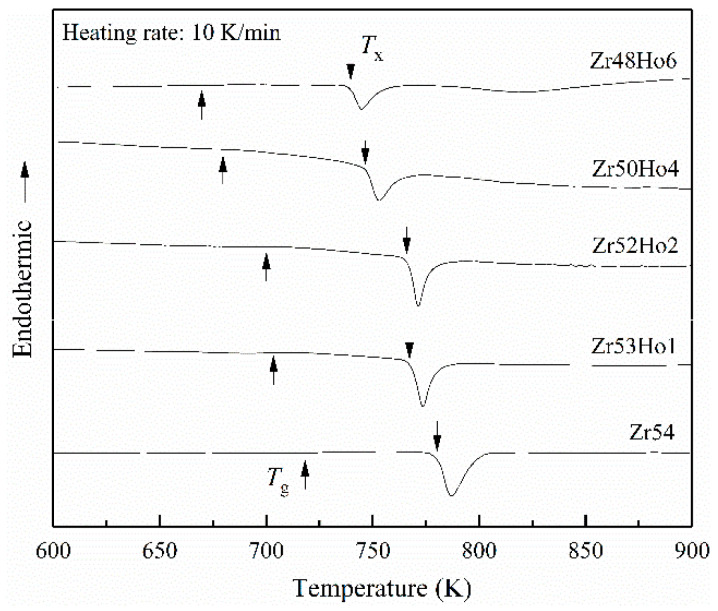
DSC curves of Zr_54−*x*_Cu_29_Ni_7_Al_10_Ho*_x_* (*x* = 0, 1, 2, 4, 6) BMGs.

**Figure 3 materials-15-02516-f003:**
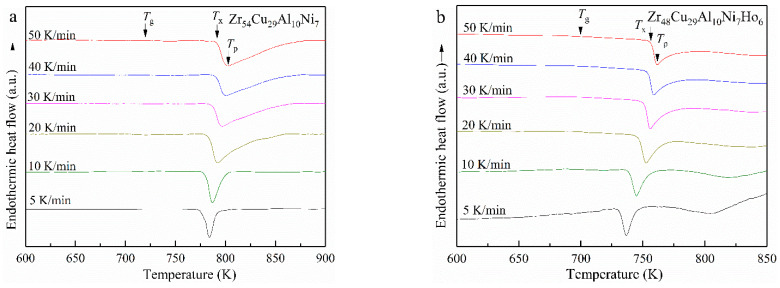
DSC results of Zr54 (**a**) and Zr48Ho6 (**b**) BMGs at various heating rates.

**Figure 4 materials-15-02516-f004:**
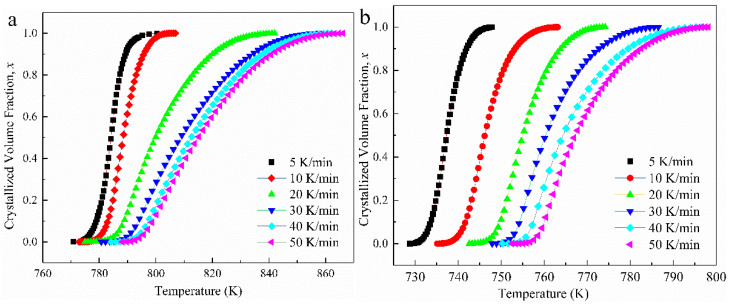
Crystallized fraction *x* corresponding temperature at different heating rates (**a**) Zr_54_ BMG and (**b**) Zr_48_Ho_6_ BMG.

**Figure 5 materials-15-02516-f005:**
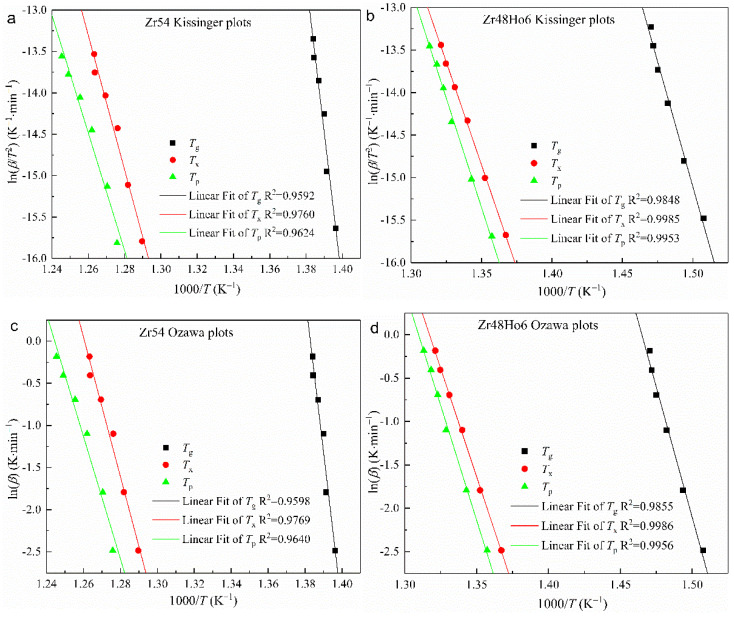
Kissinger and Ozawa plots of (**a**,**c**) Zr54 BMG and (**b**,**d**) Zr48Ho6 BMG.

**Figure 6 materials-15-02516-f006:**
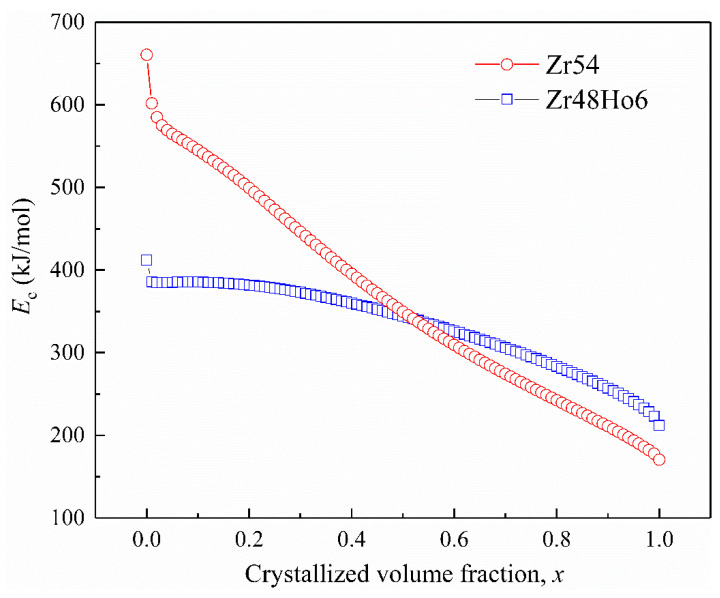
Local activation energies corresponding to crystallized fraction for Zr54 and Zr48Ho6 BMGs.

**Figure 7 materials-15-02516-f007:**
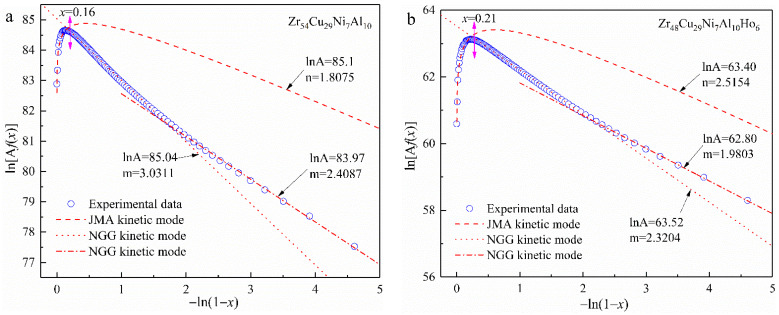
Curves representing crystallization mechanisms of (**a**) Zr54 BMG and (**b**) Zr48Ho6 BMG.

**Figure 8 materials-15-02516-f008:**
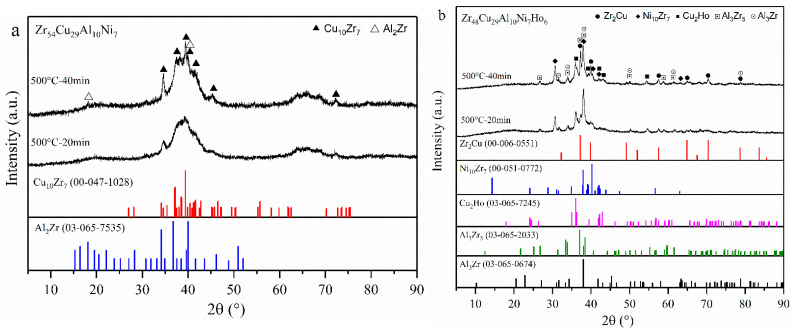
XRD patterns of (**a**) Zr54 BMG and (**b**) Zr48Ho6 BMG annealed at 500 °C for various times.

**Figure 9 materials-15-02516-f009:**
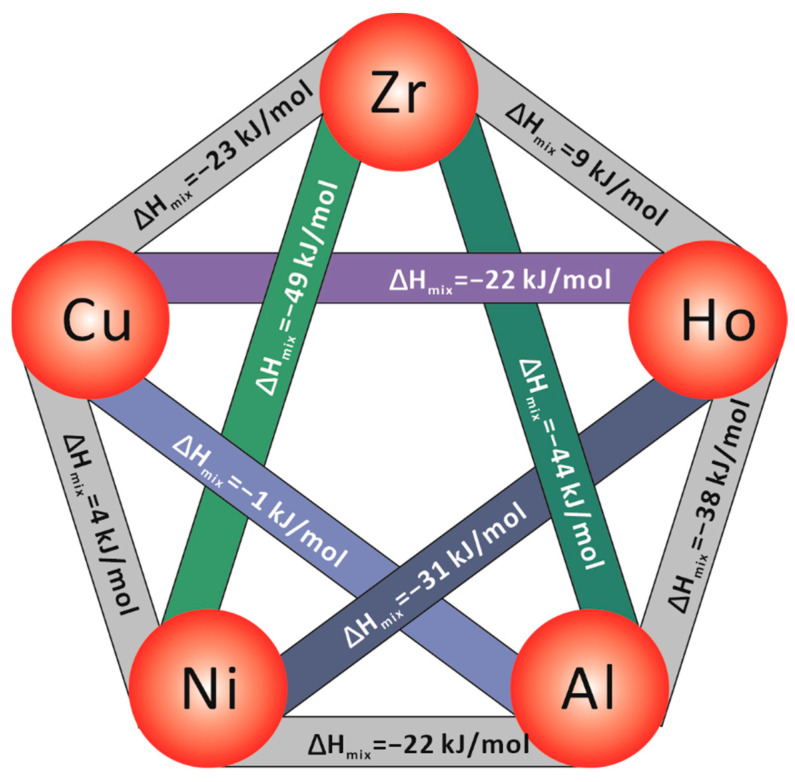
Mixing enthalpy between constituent elements in the Zr-Cu-Ni-Al-Ho system.

**Figure 10 materials-15-02516-f010:**
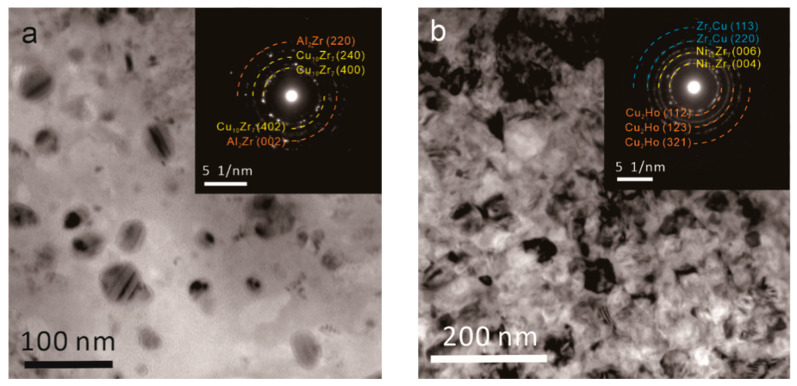
TEM images and the corresponding SAED results of (**a**) Zr54 BMG and (**b**) Zr48Ho6 BMG annealed at 500 °C for 20 min.

**Table 1 materials-15-02516-t001:** Thermal parameters of Zr_54-*x*_Cu_29_Ni_7_Al_10_Ho*_x_* (*x* = 0, 1, 2, 4, 6) BMGs.

Composition (in at.%)	*T_g_* (K)	*T_x_* (K)	Δ*T_x_* (K)
Zr_54_Cu_29_Ni_7_Al_10_	718.7	780.1	61.4
Zr_53_Cu_29_Ni_7_Al_10_Ho_1_	703.5	767.2	63.7
Zr_52_Cu_29_Ni_7_Al_10_Ho_2_	700.1	765.3	65.2
Zr_50_Cu_29_Ni_7_Al_10_Ho_4_	679.7	746.2	66.5
Zr_48_Cu_29_Ni_7_Al_10_Ho_6_	669.5	739.4	69.9

**Table 2 materials-15-02516-t002:** Characteristic temperatures of Zr54 BMG and Zr48Ho6 BMG at various heating rates.

Heating Rate (K/min)	Zr_54_Cu_29_Ni_7_Al_10_				Zr_48_Cu_29_Ni_7_Al_10_Ho_6_			
	*T_g_* (K)	*T_x_* (K)	*T_p_* (K)	Δ*T_x_* (K)	*T_g_* (K)	*T_x_* (K)	*T_p_* (K)	Δ*T_x_* (K)
5	716.2	775.4	783.79	59.2	663.3	731.4	736.71	68.1
10	718.7	780.1	787.1	61.4	669.5	739.4	744.65	69.9
20	719.4	783.6	792.39	64.2	674.65	746.25	752.59	71.6
30	721	787.7	796.47	66.7	677.97	751.27	755.87	73.3
40	722.4	791.4	800.57	69	679.39	754.89	758.51	75.5
50	722.5	791.6	802.93	69.1	680.04	756.84	761.51	76.8

**Table 3 materials-15-02516-t003:** Activation energies obtained by two equations for Zr54 BMG and Zr48Ho6 BMG.

Equations	Zr_54_Cu_29_Ni_7_Al_10_			Zr_48_Cu_29_Ni_7_Al_10_Ho_6_		
	*E_g_* (kJ/mol)	*E_x_* (kJ/mol)	*E_p_* (kJ/mol)	*E_g_* (kJ/mol)	*E_x_* (kJ/mol)	*E_p_* (kJ/mol)
Kissinger	1525.07	675.67	590.68	488.87	402.43	427.39
Ozawa	1537.03	688.71	603.87	500.04	414.80	439.84

**Table 4 materials-15-02516-t004:** Theoretical kinetic modes considered.

Model	*f*(*x*)	Label
Johnson–Mehl–Avrami (JMA)	*n*(1 − *x*)[−ln(1 − *x*)]^(*n* − 1)/*n*^	*n*
Normal grain growth (NGG)	(1 − *x*)*^m^* ^+ 1^	*m*

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
