# Peer review of "Effect of Ho Addition on the Glass-Forming Ability and Crystallization Behaviors of Zr54Cu29Al10Ni7 Bulk Metallic Glass"

_materials, 2022, doi:10.3390/ma15072516_

Round 1

Reviewer 1 Report

The authors investigate the effect of Ho addition on the glass-forming ability and crystallization of Zr-Cu-Al-Ni Bulk metallic glass. The authors employ several experimental techniques to evaluate the BMG crystalline structure, characteristic temperatures and activation energy of crystallization. The presented results are interesting and worth publication after my queries and recommendations are addressed.

1) The authors include Ho as substituting Zr. How was it determined that Ho occupies Zr sites on the crystalline structure? If not, Which site does Ho occupy on the crystal structure.

2) On Figure 2 the authors present DSC data showing Tg and Tx. How Tg is evaluated since it does not show any visible impact on DSC measurements?

3) Does Tg and Tx present Hysteresis? I mean, if the measurements were taken on cooling any change on those temperatures are expected?

4) Why the volume fraction of crystallization is calculated as mention on the manuscript? Please clarify it and include some references.

In conclusion, after my recommendation are adressed the manuscript will be suitable for publication on Materials.

Author Response

Response to Reviewer 1 Comments

Point 1: The authors include Ho as substituting Zr. How was it determined that Ho occupies Zr sites on the crystalline structure? If not, Which site does Ho occupy on the crystal structure.

Response 1: Thank you very much for the reviewer’s professional advice. As the amorphous structure of alloys, the distributions of all elements are disorder in the long range. The addition of Ho just decreased the content of Zr. However, the short-range order should be changed by the addition of Ho. The structure of short-range order is generally investigated by synchrotron radiation and molecular dynamics simulation.

Point 2: On Figure 2 the authors present DSC data showing Tg and Tx. How Tg is evaluated since it does not show any visible impact on DSC measurements?

Response 2: Thank you very much for the reviewer’s professional advice. It can be seen in Fig. 2 that Tg decreased with the increased content of Ho, indicating the decreasing critical temperature of glass transition for metallic glasses. The decreasing Tg implies the lower temperature for the relaxation of atoms in the metallic glasses.

Point 3: Does Tg and Tx present Hysteresis? I mean, if the measurements were taken on cooling any change on those temperatures are expected?

Response 3: Thank you very much for the reviewer’s professional advice. The hysteresis phenomenon will be found in the DSC curves at high heating rates. So, we selected the heating rate of 10 K/min to measure the characteristic temperatures. However, the hysteresis of characteristic temperatures at higher heating rates did not affect the contrast on crystallization kinetics of different metallic glasses. The amorphous structure is formed by quenching in the preparation and is nonequilibrium situation, which is the reason of crystallization at the elevating temperature. So, the exothermic peak of crystallization can only be measured during raising temperature. In the cooling process, the exothermic peak of solidification will be measured. Because the cooling rate in DSC is not fast enough, the solidified alloy is not amorphous and the corresponding characteristic temperatures will not be detected.

Point 4: Why the volume fraction of crystallization is calculated as mention on the manuscript? Please clarify it and include some references.

Response 4: Thank you very much for the reviewer’s professional advice. The calculation of crystallization fraction is of vital importance. As the nucleation and growth of metallic glass are complicated, the activation energy over the total crystallization process is not fixed. The crystallization fraction and corresponding activation energy was calculated to evaluate the whole crystallization process, which could describe the nucleation and development of crystallization. The corresponding references are cited in the manuscript.

Reviewer 2 Report

The manuscript Effect of Ho addition on the glass-forming ability and behaviors of Zr54Cu29Al10Ni7 bulk metallic glass studies general properties of Zr-based BMG with Ho addition. The experimental results are reported in detail and well presented. In general, the reviewer thinks the manuscript is good. However, the following comments should be considered before making a decision.

  1. Line 23-24 is not misleading. The lack of plasticity is the common problem of metallic glasses. Among commonly used MG, Zr-based MG has relatively high plasticity.
  2. Line 40-41, what do you mean by ”less additive amount”? The choice of Ho addition should be further explained.
  3. Line 62, details about TEM experiment should be included including sample preparation, TEM equipment information.
  4. Table 2, Zr54Cu29Ni7Al10Ho6 should be Zr48Cu29Ni7Al10Ho6
  5. In section 3.6, the reasoning of crystallization is hard to follow. For example, line 201-203, why "reduced Cu atoms combining with Zr atoms" then causes "the precipitation of CuZr2, 203 Ni10Zr7 and Al3Zr5"? Line 206-207, mixing enthalpy between Cu-Ho is neither the largest nor the smallest, why Ho combined preferentially with Cu to form precipitates?
  6. In the discussion part, a comparison between the alloying effect of Ho and other rare earth element is expected.
  7. Line 250, point (5) should be deleted.

Author Response

Response to Reviewer 2 Comments

Point 1: 1. Line 23-24 is not misleading. The lack of plasticity is the common problem of metallic glasses. Among commonly used MG, Zr-based MG has relatively high plasticity.

Response 1: Thank you very much for the reviewer’s professional advice. Our expression is misleading. The plasticity of Zr-based MG is excellent among commonly used MGs. We want to express the necessary of improvement in plasticity for commercial application. We have corrected the corresponding expression in the revised manuscript.

Point 2: Line 40-41, what do you mean by ”less additive amount”? The choice of Ho addition should be further explained.

Response 2: Thank you very much for the reviewer’s professional advice. We have corrected the expression and added the corresponding explanation in revised manuscript.

Point 3: Line 62, details about TEM experiment should be included including sample preparation, TEM equipment information.

Response 3: Thank you very much for the reviewer’s professional advice. We have added the equipment information and experimental details of TEM in the revised manuscript.

Point 4: Table 2, Zr54Cu29Ni7Al10Ho6 should be Zr48Cu29Ni7Al10Ho6

Response 4: Thank you very much for the reviewer’s kind remind. We have corrected the mistakes in the revised manuscript.

Point 5: In section 3.6, the reasoning of crystallization is hard to follow. For example, line 201-203, why "reduced Cu atoms combining with Zr atoms" then causes "the precipitation of CuZr2, 203 Ni10Zr7 and Al3Zr5"? Line 206-207, mixing enthalpy between Cu-Ho is neither the largest nor the smallest, why Ho combined preferentially with Cu to form precipitates?

Response 5: Thank you very much for the reviewer’s professional advice. Our expression is not clear in this section. The Cu atom was priority in combination with Ho to form Cu2Ho, leading the decreasing disordered Cu atoms in the alloy. The remaining Cu to Zr ratio was reduced and resulted in the precipitation of CuZr2 rather than Cu10Zr7. CuZr2 consumed more Zr atoms and leaded the formation of compounds with low Zr content, such as Ni10Zr7 and Al3Zr5.

Although the mixing enthalpies of Ni-Ho and Al-Ho are more negative than that of Cu-Ho, but less negative than those of Ni-Zr, Ni-Al and Zr-Al. Ni will combine preferentially with Zr and Al, and Al will combine preferentially with Zr and Ni, rather than Ho. Therefore, Ho combined preferentially with Cu to form precipitates Cu2Ho, which is in good agreement with the XRD results.

Point 6: In the discussion part, a comparison between the alloying effect of Ho and other rare earth element is expected.

Response 6: Thank you very much for the reviewer’s professional advice. Overall, the alloying effects of Ho on GFA and crystallization are similar to other rare earth elements, including the improved GFA and reduced activation energy of crystallization. The addition of rare earth made the MG much easier to form rare-earth compounds and nucleation. Compared with the addition of Y, the alloying of Ho leaded to the different precipitates Cu2Ho, which resulted in the difference of crystallized products and corresponding proportion. And then, the nanocrystalline alloy with different structures and properties can be obtained. We also added this comparison in the revised manuscript.

Point 7: Line 250, point (5) should be deleted.

Response 7: Thank you very much for the reviewer’s kind remind. We have deleted it in the revised manuscript.

Round 2

Reviewer 2 Report

Point 4, The mistake inside Table 2 hasn't been corrected: Zr54Cu29Ni7Al10Ho6 should be Zr48Cu29Ni7Al10Ho6.

Author Response

Thank you very much for the reviewer’s kind remind again. We have corrected the mistakes in the revised manuscript.